# Engineering Properties of Waste Sawdust-Based Lightweight Alkali-Activated Concrete: Experimental Assessment and Numerical Prediction

**DOI:** 10.3390/ma13235490

**Published:** 2020-12-02

**Authors:** Hisham Alabduljabbar, Ghasan Fahim Huseien, Abdul Rahman Mohd Sam, Rayed Alyouef, Hassan Amer Algaifi, Abdulaziz Alaskar

**Affiliations:** 1Department of Civil Engineering, College of Engineering, Prince Sattam bin Abdulaziz University, Alkharj 11942, Saudi Arabia; h.alabduljabbar@psau.edu.sa; 2Construction Research Centre, Institute for Smart Infrastructure and Innovative Construction, School of Civil Engineering, Faculty of Engineering, Universiti Teknologi Malaysia, Johor Bahru 81310, Malaysia; abdrahman@utm.my; 3Faculty of Civil and Environmental Engineering, Universiti Tun Hussein Onn Malaysia, Parit Raja 86400, Malaysia; enghas78@gmail.com; 4Department of Civil Engineering, College of Engineering, King Saud University, Riyadh 11362, Saudi Arabia; abalaskar@ksu.edu.as

**Keywords:** lightweight concrete, alkali-activated, waste sawdust, predicted engineering properties, sustainability

## Abstract

Alkali activated concretes have emerged as a prospective alternative to conventional concrete wherein diverse waste materials have been converted as valuable spin-offs. This paper presents a wide experimental study on the sustainability of employing waste sawdust as a fine/coarse aggregate replacement incorporating fly ash (FA) and granulated blast furnace slag (GBFS) to make high-performance cement-free lightweight concretes. Waste sawdust was replaced with aggregate at 0, 25, 50, 75, and 100 vol% incorporating alkali binder, including 70% FA and 30% GBFS. The blend was activated using a low sodium hydroxide concentration (2 M). The acoustic, thermal, and predicted engineering properties of concretes were evaluated, and the life cycle of various mixtures were calculated to investigate the sustainability of concrete. Besides this, by using the available experimental test database, an optimized Artificial Neural Network (ANN) was developed to estimate the mechanical properties of the designed alkali-activated mortar mixes depending on each sawdust volume percentage. Based on the findings, it was found that the sound absorption and reduction in thermal conductivity were enhanced with increasing sawdust contents. The compressive strengths of the specimens were found to be influenced by the sawdust content and the strength dropped from 65 to 48 MPa with the corresponding increase in the sawdust levels from 0% up to 100%. The results also showed that the emissions of carbon dioxide, energy utilization, and outlay tended to drop with an increase in the amount of sawdust and show more the lightweight concrete to be more sustainable for construction applications.

## 1. Introduction

Sawdust is a well-known agriculture and by-product waste material resulting from the wood industry. It is generated as a waste material when timbers are mechanically milled into different sizes and shapes. Many environmental problems are caused by sawdust wastes, wherein the scarcity of space for land fill is a major concern and a severe threat to developed nations. The excessive sawdust wastes that are accumulated due to the activities of factories, mills, and houses are ever growing annually. It is estimated that the annual generation of wood waste in the United States of America, Germany, the United Kingdom, and Australia is around 64, 8.8, 4.6, and 4.5 million tonnes per years, respectively, and more than 40% of these amounts are not recycled [1,2,3,4]. The high percentage of non-recycled wood wastes shows the deficiency of sufficient recycling procedures and strategies. Thus, it is vital to recycle wood wastes on a daily basis and utilize them effectively in cement-based composites/concretes to guarantee their harmless discarding as an environmental remedy.

Currently, researchers are facing a great challenge because of the constant increase in the demands of high-performance lightweight concretes (LWCs) as construction materials, where the manufacturing of novel construction materials from recycled industrial wastes has become a strategy. In this view, the advancement of LWCs via the use of sawdust wastes as lightweight aggregates is evaluated. The functions of sawdust in cements/concretes have been assessed by several researchers, and it has been used to make lightweight concretes in the past [5]. The thermal traits of sawdust-based cement composite have been reported [6], wherein its inclusion in the concrete matrix was found to significantly reduce the thermal conductivity by up to 20% compared to that of normal concrete (0% sawdust). Such a considerable decrease in the conductivity values was ascribed to the lowering in density and increased porosity of the lightweight concrete composites modified by sawdust wastes. Oyedepo et al. [7] used sawdust wastes as a substitute for fine aggregates (natural) at different contents from 0% up to 100% in standard heavyweight concretes, and showed that a ratio of more than 25% substitute to natural aggregates can negatively influence the concrete’s strength properties and density. Other researchers have also made comparable observations when sawdust was used in concrete at various levels (10%, 20%, 30%, and 40%) as a substitute for sand. It was suggested that an amount of sawdust at up to 10% substitution for sand could produce a better density and mechanical strength of concrete [8]. Boob [9] also used sawdust as a substitute for fine aggregates (0–15%) in concrete. Mageswari and Vidivelli [10] showed that sawdust ash as an agent to replace natural sand may be an appropriate choice for fine aggregates in concretes. It can significantly decrease the sawdust waste clearance problem and concurrently allow the conservation of natural fine aggregates. The authors found that concrete including sawdust possessed unique characteristics and presented better outcomes for the thermal and mechanical characteristics of the cement-based composite, making it economical compared to various other materials in the construction sector.

Lately, several products such as geopolymer and alkali-activated materials have been introduced as alternatives to conventional concrete and have emerged as constructional materials with lower CO_2_ footprints [11,12,13,14,15,16,17]. Alkali-activated pastes/mortars/concretes are inorganic polymers based on calcium (CaO) and alumina-silicates (ASs) activated with alkaline activator solution. These are prepared from pozzolanic compounds via the alkali activation of NaOH and sodium silicates (NaSi) [18,19]. These binders obtained using alkali activation showed eco-friendliness due to the need for a modest quantity of energy in their fabrication process [20,21]. Following alkali activation, various solid wastes from different industries that contain Si, Al, and/or Ca, including fly ash (FA), palm oil fuel ash (POFA), metakaolin, and granulated blast furnace slag (GBFS), have been used to make mortars/concretes [22,23,24].

Several researchers [25,26] have observed that FA containing a high amount of CaO is also a proper resource material for producing high-performance geopolymer mortars and concretes. It was shown that the mixture of geopolymer prepared with FA class C (high CaO) become curable at room temperature because of the CaO-mediated reaction. Nevertheless, the geopolymerization of FA class C in the absence of additive was found to be very sluggish at ambient temperatures [27], achieving a low strength. Yet, the usage of materials containing a high amount of CaO, including Ordinary Portland Cement (OPC), to enhance the strength of high-CaO FA-based geopolymer remains prospective [28]. Besides the generation of calcium-silicate-hydrate (C–S–H) and calcium-aluminium-silicate-hydrate (C–A–S–H), the produced heat and water from the OPC-mediated reaction can help the geopolymerization process and thereby the development of strength enhancement [29]. By incorporating OPC and curing at 25 °C, high-Ca FA-based geopolymer mortars with a compressive strength (CS) of 65 MPa were produced [30].

Amorphous GBFS, being one of the most popular industrial wastes, has been widely used to enhance normal concrete durability or fabricate cement-free mortars/concretes because of its excess contents of Al_2_O_3_, CaO, and SiO_2_ in its chemical composition [31,32,33,34,35]. In alkaline media, GBFS shows both binding and pozzolanic properties [36]. Many investigations have detected [37] that the generation of excess Ca because of the addition of GBFS in FA geopolymer is accountable for the enhancement of the strength characteristics as well as the microstructure of the material. To evaluate the effectiveness of GBFS including FA as a geopolymer binder, the FA/GBFS ratio was widely varied together with the types, concentrations, and compositions of the activator in the mixture to produce them [38,39]. The inclusion of a high amount of Ca containing only 4% GBFS was found to enhance the strength of geopolymer [38]. Ismail et al. [40] evaluated the CS and hydration product of the FA and GBFS pastes and showed an enhancement in the CS of up to 50 MPa at the curing age of 28 days. An elevation in the FA to GBFS ratio of as much as 1.0 was used and it was activated by 10 M of NH solution, before curing at 25 °C was carried out. According to Ismail et al. [41], an early-age compressive strengths (CSs) of the FA/GBFS composite activated by the NH/NS may increase considerably with a minute quantity of hydrated lime. This FA geopolymer blended with slag exhibited an excellent mechanical and durability performance [42]. Previous studies have attempted to fabricate eco-friendly high-performance LWCs, cement-free concretes, and alkali-activated geopolymers, where the primary focus was to achieve improved strength and durability characteristics.

## 2. Research Significance

A comprehensive literature overview revealed that the potential usage of sawdust wastes for developing alkali-activated LWCs for the sustainable performance has not been widely explored yet. This work reports the effects of sawdust waste substitution for natural aggregates on the sustainability characteristics of LWCs with alkali activation containing FA and GBFS. These mixtures were made at changing levels of sawdust, including 70% FA, 30% GBFS, and alkali-activated solution to find the feasibility of recycling industrial wastes and transforming them into environmentally responsive, long-lasting, and sustainable lightweight concrete. Thus, natural aggregates were replaced by different levels of sawdust wastes (0%, 25%, 50%, 75%, and 100%) at a realistic working level with the appropriate physical conditions to make the alkaline solution-activated LWCs. All the synthesized specimens were analyzed by various measurements to evaluate the fresh, mechanical, and durability properties for obtaining an optimal composition.

## 3. Experimental Details

### 3.1. Materials

Furnace slag (GBFS of an off-white color) of a high purity was collected from a Malaysian industry (Ipoh, Malaysia) and utilized without further purification to produce cement-free binder. It was different from other supplementary components, with both cementitious and pozzolanic properties. It is obtained from the hydraulic chemical reactions upon mixing water. The X-ray fluorescence (XRF, HORIBA, Singapore, Singapore) spectra test of the slag showed the presence of Ca (51.8%), silicate (30.8%), and Al (10.9%). Low-level Ca containing FA (alumina-silicate material with a grey appearance) was obtained from a Malaysian power station (Tanjung bin, Johor, Malaysia) for producing the proposed AAMs. It fulfilled the requisites of the ASTM C618 for FA class F and contained Ca (5.2%), silicate (57.2%), and Al_2_O_3_ (28.8%). The particle median for the FA and slag (achieved by a particle size analyzer) was, respectively, 10 and 12.8 µm. The physical characteristics of both binder materials (GBFS and FA) were analyzed using the Brunauer Emmett Teller (BET, JEOL, Kuala Lumpur, Malaysia)) test with specific surface area (18.1 m^2^/g for FA and 13.6 m^2^/g for GBFS) calculations. 

Figure 1 presents the X-ray diffraction (XRD, Rigoku, Singapore, Singapore) pattern of GBFS and FA. The observed intense XRD peaks of FA at 2θ = 16–30° were due to the existence of polycrystalline silica and Al_2_O_3_. However, the prominent peaks at other angles were due to the existence of quartz and mullite crystallites. The absence of any sharp peak of GBFS verified its amorphous nature. The presence of silica and Ca peaks played an important role in the composition of GBFS and was beneficial for the AAM production. Conversely, the incorporation of FA was required to overcome the low level of Al_2_O_3_ (10.49%) in the slag.

Natural river sand was used as the fine aggregate to produce the control concrete samples. Following the ASTM C117 protocol, first the sand was washed in water to eliminate the silts and impurities [43], followed by the oven drying at 60 °C for 24 h to remove the moisture. The obtained clean sand fulfilled the ASTM C33–33M requisites [44]. The fineness modulus, specific gravity, and highest particle size of the prepared sand were 2.9, 2.6, and 2.36 mm, respectively. Crushed garnet stone obtained from a quarry was used as a coarse aggregate in the sample preparation process. In producing conventional concrete, the size of the coarse aggregate plays an important role in ensuring that a good performance of concrete could be achieved. Therefore, the highest size of coarse aggregates was limited to below 8 mm. 

The sawdust wastes (No. 6013) were obtained (Figure 2) from the Malaysian (Syarikat Kilang Papan Chong Wah Sdn Bhd., Johor, Malaysia) wood industry. This local agro-waste ensured the acquirement from a single resource (density of 174 kg/m^3^ and maximum size of 2.36 mm) for the fine aggregate use. Sawdust with a density of 182 kg/m^3^ and a maximum size of 6 mm was utilized as a coarse aggregate to prepare the LWC. The main attributes of the sawdust include the chemical composition and the loss of ignition (LOI), as shown in Table 1. The primary constituent of the sawdust was cellulose (87% of the total mass) and low amounts of CaO and Al_2_O_3_. The LOI percentage of the sawdust from the total mass was found to be 4.76%.

The solution (S) for the alkaline activation was made of sodium hydroxide (NH) and sodium silicate (NS). It was used to dissolve the alumina-silicate from FA and GBFS. Analytical-grade NH (98% purity) pellets were dissolved in water to prepare a solution of 13.7% of Na_2_O and 86.3% of H_2_O (2 M). A high-purity NS mixture was prepared using SiO_2_ (29.5 wt. %), Na_2_O (14.70 wt.%), and H_2_O (55.80 wt.%). The obtained NH solution (2 M) was first stored for 24 h at room temperature and later mixed with NS solution to obtain the final alkali solution with a modulus (Ms of SiO_2_:Na_2_O) of 1.21. The ratio of the NS to NH for all the alkali solutions was kept constant at 0.75.

### 3.2. Mix Designs of Prepared Concretes

For all the LWC specimens, the values of the alkaline solution to binder ratio (S:B) and the binder content were fixed with 0.40 and 450 kg/m^3^, respectively. Waste products such as FA and GBFS were utilized for fabricating LWC mixes with constant amounts of 70% and 30%, respectively, as sources of SiO_2_, Al_2_O_3_, and CaO. A blend containing 100% natural aggregates (sand and gravels) was made and regarded as the control sample (Table 2). The molarity of the NH, NS to NH, and alkaline solution modulus (Ms) was fixed for all concrete mixtures. The influence of various contents of fine and coarse sawdust as a natural aggregate replacement on the LWC design is shown in Table 2. Four replacement contents were used to evaluate the effects of sawdust waste on the proposed concrete’s weight, strength, and geopolymerization process.

### 3.3. Fresh and Hardened Concretes Tests Program

Prior to the mixing and casting, the internal surface of the molds was greased with engine oil to make the de-molding process easy. A homogeneous alkali solution composed of NH and NS was cooled at ambient atmosphere and then used for the concrete preparation. Uniform mixtures of fine/coarse aggregates were made by blending FA/GBFS for approximately 4 min at dry conditions. Next, the prepared mixes were alkali-activated. The whole concrete matrix was mixed for 4 min once more via a machine controlled at an average speed. The achieved fresh green concretes were cast within the molds in three layers, wherein every layer was strengthened via the vibration table for 30 s to remove air voids. Upon the completion of the casting process, the casted concretes were cured at 27 ± 1.5 °C (for 24 h at relative humidity of 75%). Finally, the concrete mixes were de-molded and stored under identical settings for further testing and analyses.

Following the ASTM C143 and C191 protocols, the slump and setting time values were measured, respectively. The CS measurements were conducted in cubic-shaped molds of size 100 × 100 × 100 mm which were adequately cured for 1, 3, 7, 28, 56, and 90 days following the ASTM C579 specification. These CS tests were performed following the ASTM C109-109M standard, where three sets of samples were analyzed at each curing age. A load at constant rate (2.5 kN/s) was subjected to test the failure of these specimens. Since the machine has inbuilt configurations, the density and CS were generated automatically depending on the imputed specimen weight and dimensions. A prism-shaped sample with the dimensions of 100 mm × 100 mm × 500 mm were cast for flexural strength (FS) and drying shrinkage (DS) tests following the ASTM C78 and C157 stipulations, respectively. The average readings of the three tested concrete mixes at the curing ages of 3, 7, 14, 21, 28, 56, and 90 days were considered to assess the DS value of each mix. According to the ASTM C496 standard, cylindrical-shaped specimens (diameter of 100 mm and depth of 200 mm) were prepared for a splitting tensile strength (STS) evaluation. A water absorption (WA) test was performed following the ASTM C642 specification, wherein mixes of size 100 mm × 100 mm × 100 mm were molded. The specimens were immersed in water at 27 °C for 24 h after they matured. Later, these specimens were suspended and completely submerged in water to measure their weight (Ms). Subsequent to the saturation, all the specimens were dried in a ventilated oven at 105 °C for over 24 h then weighed (Md). The WA of the proposed LWCs was calculated from the average value of the three samples via the relation:(1)$$\mathrm{WA}\left(\%\right)=\frac{\mathrm{Ms}-\mathrm{Md}}{\mathrm{Md}}\;\times 100$$

### 3.4. Artificial Neural Network (ANN) Model

In this work, the ANN model was utilized to explain the CS of the alkali-activated concretes for obtaining the optimum values of the affecting parameters. In addition, it was intended to reduce both time and cost. The model was inspired by a natural human process. The developed model consists of three layers, as shown in Figure 3. The first layer—namely, input layer (I)—contains five neurons (parameters), which are represented by molarity, NS/NH, yeast solution to binder, GBFS/FA, and time. Then, fourteen neurons of the hidden layer (H) were used to achieve the best performing model. Meanwhile, one neuron in the third layer was used to reflect the predicted compressive strength—namely, the output layer (O).

A total of 144 experimental works were utilized to construct the proposed ANN model in MATLB. In particular, the feed-forward back-propagation network architecture was created using a newff function. In addition, the sigmoid function was adopted to map the input with the target output, as shown in Equation (2).
(2)$$f\left(x\right)=\;\frac{1}{1+e^{-x}}$$

As much as 75% of the experimental data was used for training using the Levenberg–Marquardt (LM) algorithm in order to minimize error. Meanwhile, 15% and 10% of the experimentally measured values were utilized to test and validate the proposed model, respectively. Equation (3) was used to convert the experimental data values to normalized ones. The normalized values were ranged between 0.1 and 0.9, aiming to avoid any scaling impact. Here, *Xi* is the input or output value, while *Xmax* and *Xmin* are the corresponding highest and lowest values. Furthermore, the performance of the proposed model was evaluated based on both the coefficient of correlation (*R*^2^) and error, with a performance goal of 0.01 and learning rate of 0.2.
(3)$$Xnorm=0.8\;\times \left(\frac{Xi-Xmin}{Xmax-Xmin}\right)-0.1$$

The correlation coefficient (*R^2^*) was also taken into account as a statical estimator. In particular, it was utilized to evaluate the strength of the results. In addition, *R^2^* is able to provide insight into the degree of fitting between the network output and the collected experimental data, as expressed in Equation (4). Accordingly, *Yactual* was the experimental result of the concrete strength and *Ymodel* was the predicted concrete strength from the model. In addition, the average value of the predicted results was termed *Ymodel mean*, whereas the number of experimental runs was represented by N. Moreover, the best fitting of the actual CS of the alkali-activated and the predicted results was accrued by increasing the value of the correlation coefficient, wherein the values usually ranged from 0–1.
(4)$$R^{2}=\frac{\sum_{i=1}^{n}{\left(Yacual\;i-Y\;model\;mean\right)}^{2}-\;\sum_{i=1}^{n}\left(Y\;model-Y\;actual\;i\right)\;}{\sum_{i=1}^{n}\left(Yacual\;i-Y\;model\;mean\right)}$$

### 3.5. Sound Absorption 

The acoustic energy absorption, reflection, and dissipation capacity of the material were obtained from the sound absorption measurements. In accordance to the ASTM E1050 stipulation, the two-microphone transfer-function (impedance tube) method was used to determine the impedance and absorption of the acoustic specimens. This method is intended for measuring the absorption coefficient and the particular acoustic impedance of sound-absorbing materials that are circular-cut in small samples, normally in the 100 to 6000 Hz frequency range (Figure 4).

### 3.6. Heat Transfer Measurement

In developed countries, buildings are large consumers of energy and saving energy is the main concern. The energy consumed in buildings can be saved effectively by increasing their thermal insulation, which is vital for countries with hot and cold climates and a high energy demand. Thermal insulation is required to decrease the total energy usage in buildings and add to unusual regenerative energy resources for sustainability. The heat transfer was measured for the cylindrical specimen of diameter 150 mm and height 300 mm. After 28 days of casting, the dried surface of the specimen was covered by a plastic sheet to prevent the excess entry of water. A PVC pipe (20 mm of diameter) was used to protect the thermocouple from unexpected impacts. All the samples were put in a water container at 34 °C. Then, the temperature of the water was slowly increased up to 100 °C, wherein the first measurement was conducted. Next the heater was turned on to record the interior temperature of the specimen using a K thermocouple, data loggers, and computers. During the immersion of the sawdust concretes in water, the temperature of the heater was increased, thereby increasing the water volume. Such a rising temperature of water was recorded at close intervals in the first 24 h up to 100 °C. However, the transferred heat was measured later and rather frequently until the water temperature came down to boiling point.

### 3.7. Environmental and Economic Benefits

For lightweight concretes to be a practicable product like the traditional one, they must have a lower or comparable cost for the user, considerably improved function or ease of manufacture, or other sustainability benefits. To compare the sawdust-based concretes with the conventional one in terms of sustainability, three headline metrics were selected, such as the carbon dioxide emissions, the usage of energy (direct fuel consumption), and the total production cost. These matrices were used to argue in favor or against the usage of sawdust-based lightweight concretes. However, other key factors that have considerable roles are technical presentation, leaching, water consumption, contents of harmful materials, release of other greenhouse gases, and waste volume amount. These metrics can be avoided using sawdust in an activated alkaline solution or sand and gravel-based concretes. Actually, the chosen 3 metrics are used to quantify the development of alkaline solution-activated lightweight concretes in the industry at early stages.

The CO_2_ emission, energy usage, and cost were derived via the life cycle approach. This evaluation implied the requirement of feed stocks for manufacturing the aggregates and the related transport cost. It allowed a valid comparison among sawdust, sand, and gravel, wherein the production impact was unable to provide a complete depiction of the energy demand and the CO_2_ emission from the feedstock. These do not include factors such as the mixing, laying, and curing of the alkaline solution-activated concrete and the operation life span emission, assuming them to be alike for every product. This strategy is comparable in terms of the life cycle impacts rather than the absolute impacts. As an effective method for comparable products, it lowers the required evaluation time.

Following the life cycle of every material, the outlay, the amount of CO_2_ release, and the energy requirements were estimated. The life cycle of fine and coarse sawdust includes the collection and transport stages. The collection cost of sawdust wastes from the factory was negligible. The distance for the transportation of each substance was added in the life cycle estimate. The distance for the gravel transportation was longer (60 km) compared to that of sand (49 km) and sawdust (5 km). The fuel cost of transports, including the kinds of trucks, volume, speed, and cost for 1 tone/km, were same for every type of material. Table 3 lists the machineries and materials needed for life cycle calculation. The total CO_2_ emission and cost of the fine/coarse aggregates were calculated relative to each material, where the total diesel consumption depended on the transportation distance (Table 3). Likewise, the total energy required for each mix was estimated depending on the diesel cost of every type of material, including crushing and transportation. Equations (5)–(7) were adopted to calculate the total CO_2_ emissions, cost, and energy consumption for each meter cubic of material. The total amounts of CO_2_ emission, outlay, and energy utilization for every material are listed in Table 4.

Total CO_2_ emission
(5)$$\sum_{i=1}^{n}mi\;\left[\left(di\times Di\times k1i\right)+\left(Ei\times k2i\right)\right]$$
where *mi* is the mass of component *i* (ton/m^3^*)*, *di* is the transport distance (km), *Di* is the diesel consumption (L/km), *k1i* denotes the CO_2_ emission for 1 Liter diesel in tonnes, *Ei* represents the total electricity consumption (kWh), and *k2i* is the CO_2_ emission for 1 kwh electricity in tonnes. 

Total energy consumption:(6)$$\sum_{i=1}^{n}mi\;\left[\left(di\times Di\times k3i\right)+\left(Ei\times k4i\right)\right]$$
where *k*3*i* is the energy consumption for 1 L diesel in GJ, *Ei* is the total electricity consumption (kWh) and *k*4*i* is the energy consumption for 1 kwh electricity in GJ.

Total cost:(7)$$\sum_{i=1}^{n}mi\;\left[\left(di\times Di\times DPi\right)+Ti+\left(Ei\times EPi\right)\right]$$
where *DPi* is the diesel cost (RM/L), Ti is the transport charge of 1 m^3^ (RM/km), and *EPi* is the electricity cost (RM/kWh).

## 4. Results and Discussion

### 4.1. Workability and Setting

Figure 5 illustrates the slump values of the prepared concretes depending on the waste sawdust replacement levels of both sand and gravel. The results of the slump reading showed that an increasing content of sawdust replacing natural aggregates reduced the workability of the prepared concretes. The value of slump was dropped from 130 to 116, 102, 91, and 74 mm with a rise in the level of replacement of sand and gravel by sawdust from 0 to 25, 50, 75, and 100%, respectively. Typically, the concrete workability was reduced with the rise in the sawdust amounts in the mixtures. Nevertheless, the influence was more prominent at higher sawdust contents (100%). Generally, the workability of the concrete was affected by the specific surface area of the sawdust and the high water demand with a high level of sawdust in the matrix. In another way, the usage of sawdust wastes as fine/coarse aggregates essentially enhanced the concretes’ texture with several other uneven and very rough fine porous particles. Therefore, it improved the inter-particle friction accountable for impeding the fresh concrete flow. For a constant solution to binder ratio, the workability of the concretes was reduced with the increase in the sawdust amount as a substitute for river sand and crushed gravel. Several researchers [45,46,47] have obtained similar trends of results for the reduction in the workability of concrete containing lightweight aggregates. The limitations associated with the workability decrease in the concrete mixes due to the utilization of sawdust wastes as substitute for natural sand can be overcome by applying super-plasticizer.

Both the initial and final setting times of the concretes produced using various amounts of the waste sawdust as fine and coarse aggregates are presented in Figure 6. Clearly, both the initial and final setting times were decreased with the increase in the sawdust levels in the concretes. For the initial setting time, the reading was decreased from 39 to 37, 34, 31, and 28 min, respectively, with an increase in the content of sawdust from 0 to 25, 50, 75, and 100%. A similar trend was found with the final setting time and reading, which was increased from 61 to 56, 53, 48, and 46 min. However, the initial and final setting time difference for each mixture was decreased with the increase in the sawdust content. Further, the difference in the setting time of each mixture is not large. The reduction in setting time with the increase in the content of sawdust addressed the high water demand of sawdust, which affected the geopolymerization process and the dissolving of alumina-silicate and calcium. An inverse relationship was observed between the sawdust waste content and the setting time of the prepared concrete. The high absorption of sawdust to alkaline solution imparted a high viscosity to the mix, which hardened rapidly. Meanwhile, the inclusion of sawdust in concrete mixtures with comparatively higher water absorptions than river sand and gravel hardened the alkaline solution-activated mixes more rapidly and decreased the setting time because of the adsorption of additional water by sawdust. Conversely, the inclusion of sawdust wastes in the alkaline system led to a reduction in the pH because of the decomposition of lignin mediated by the pH changes in the porous solution. The present outcomes are analogous to the findings of Duan et al. [48], where both the initial and final setting times were reduced for the concretes made with sawdust.

### 4.2. Hardened Density and Ultrasonic Pulse Velocity

Figure 7 depicts the hardened density values of the prepared concrete containing waste sawdust as a river sand and crushed gravel replacement at the curing age of 28 days. The densities of the prepared concretes were reduced from 2.28 to 1.98, 1.63, 1.24, and 0.89 ton·m^−3^ when the content of sawdust was increased from 0 to 25, 50, 75, and 100%, respectively. The mixture containing 100% sawdust displayed the lowest density (0.89 ton·m^−3^). In addition, the low specific gravity and porosity of the sawdust considerably influenced the densities of the prepared concretes. The present outcome is consistent with the one obtained by Memon et al. [49] for concrete including a high volume of sawdust as a coarse aggregate. Likewise, these findings contributed to improving and developing the high-performance lightweight alkali-activated concrete, which was supported by the results reported by Sales et al. [50]. They assessed the possible uses of lightweight concretes made from coarse aggregates via water treatments of sludge and sawdust. 

Figure 8 shows the influence of various sawdust amounts on the ultrasonic pulse velocity (UPV) values of the prepared concretes. An increase in the sawdust level from 0 to 25, 50, 75, and 100% caused reductions in the corresponding UPV readings of the concretes from 3.42 to 3.02, 2.79, 2.57, and 2.32 km/s at 28 days of curing age. This drop was due to the porous nature of the sawdust, which negatively influenced the densities and microstructures of the prepared concrete. It is evident that, with the increasing content of sawdust and decreasing amount of river sand/crushed gravel, the UPV of the prepared concretes was decreased, which was ascribed to the increase in the sawdust-mediated porosity. The present findings agreed well with those of a previous study [47].

### 4.3. Compressive Strength (CS)

Figure 9 shows the CS vales of the concretes containing various levels of sawdust as a substitute for sand and gravel aggregates. For each mixture, three samples were examined and the average reading was adopted. The CS of the prepared concrete was measured at the curing age of 1, 3, 7, 28, 56, and 90 days. The CS was constantly augmented with the increase in the curing age. At an early age (after 24 h), with an increase in the level of replacement of river sand and gravel by sawdust from 0 to 25, 50, 75, and 100%, the early strength dropped from 22.6 to 19.4, 18.3, 17.8, and 15.2 MPa, respectively. Beyond 28 days of the age, a comparable development was seen wherein the CS was decreased from 65.8 to 61.1, 55.7, 50.4, and 48.6 MPa with the sawdust contents increased from 0 to 25, 50, 75, and 100%, respectively. Analogous behavior was found at the late age of 90 days. where the strength loss percentage was increased with the increase in sawdust levels in the concrete matrix. It was found that the concrete achieved more than 96% compressive strength at 28 days from the total strength measured at 90 days for all mixtures, and this finding agrees with the previous finding by Ranjbar [22] and Islam [51]. However, it was found that the loss in strength was reduced with time and the percentage of loss with 100% sawdust dropped from 32.7% at an early age (1 day) to 26.1% at an age of 28 days and up. Consequently, the total reduction in CS with an increase in the sawdust content was addressed to three reasons. The first reason was that waste sawdust revealed a higher ability of water absorption than river sand and crushed gravels. Nonetheless, the varied distribution the water mixing with the concretes matrix could weaken the chemical bonds in alkali-activated paste (GBFS + FA) and aggregates. The second reason was that the shape of sawdust partials compared to those of natural aggregates made the bond between the paste and the aggregate weaker and thus there was a reduction in the concrete CS. The third reason was that the existence of organic matter led to a reduction in the aggregate-paste bonds and increased the porosity, thereby affecting the concrete CS. The fourth reason was that the substitution of the stronger substance via the weaker one and the lack of the pozzolanic action by the sawdust waste also affected negatively the strength development. This finding was consistent with the previous reports of Kanojia and Jain, S. [52], Martínez-García et al. [53], and González-Fonteboa et al. [54].

### 4.4. Flexural and Tensile Strength

The flexural strength (FS) of the LWCs were measured to evaluate their capacity for resisting the deformation when subjected to load. The FS tests of alkali-activated concretes prepared with various levels of sawdust wastes as a substitute for river sand/crushed gravel aggregates were conducted after curing for 28 days. For every mixture, the mean values of three samples were evaluated, as presented in Figure 10. The prepared specimens’ FS changes dramatically with the level of sawdust replacement of natural aggregates. It decreases from around 6.8 MPa at a 0% content to 6.2, 5.7, 5.1, and 4.9 MPa at an increasing level of sawdust to 25, 50, 75, and 100%, respectively. In terms of sawdust-based fine and coarse aggregates in alkali-activated lightweight concrete, a 100% sawdust addition exhibits the highest effects on the FS value and the concrete specimen’s loss of more than 27% flexural strength with a rise in the sawdust level from 0 to 100%. However, the FS for all mixtures achieved an acceptable strength for construction applications.

Figure 11 presented the readings of the splitting tensile strength (STS) of the LWCs obtained with various contents of the sawdust wastes as a substitute for fine/coarse aggregates. The average value of three cylinders’ concrete conducted in the evaluation of sawdust inclusion in concrete matrix. For all mixtures, the splitting tensile strength was evaluated at the curing age of 28 days. The loss in the STS was found to increase with the increase in the sawdust content, and the value of strength dropped from 4.2 to 3.9, 3.7, 3.4, and 3.0 MPa with a rise in the sawdust content from 0 to 25, 50, 75, and 100%, respectively. Similar to the reasons relating to the loss of strength (Section 3.3), the waster absorption, shape of particles, and organic substance content of sawdust led to a weak bond between the paste and sawdust as an aggregate and presented lower flexural and splitting tensile strengths compared to the control sample prepared with natural aggregates (0% sawdust) [6].

### 4.5. XRD Analysis

Figure 12 shows the XRD results of the prepared concrete containing various levels of sawdust as well as fine/coarse aggregates as a substitute. Peaks corresponding to the crystalline quartz (SiO_2_), calcium hydroxide (Ca(OH)_2_) and mullite (3Al_2_O_3_·2SiO_2_ or 2Al_2_O_3_·SiO_2_) phases were seen. These phases appeared from FA and GBFS. The intensity of the Ca(OH)_2_ diffraction peak was decreased with the increase in the sawdust amount from 0 to 25, 50, 75, and 100%, where less Portland was generated as well as a higher amount of quartz emerged to be non-reactive with 75 and 100% of sawdust (Table 5). These were produced from the chemical reaction among the amorphous fractions of FA/GBFS containing minor crystalline phases. The XRD peaks of OPC, CaCO_3_, and mullite were evidenced at 28–50°. As the sawdust contents were increased, the intensity of the XRD peak at 50.1° corresponding to the crystalline quartz phase was increased. The mullite peak at 16° for 25% sawdust also showed a lower peak intensity than the control specimen. In addition, the intensities of these peaks revealed a decreasing tendency with the increase in the sawdust contents. The peaks observed at 24° and 33.8° were assigned to the Nepheline (Na_3_KAl_4_Si_4_O_16_), where the peak intensity was reduced with the increase in the sawdust levels.

Briefly, the XRD analyses of the prepared LWCs elucidated the influence of Si, Al, and Ca on the produced C-(A)-S-H gels and the CS. The XRD results (Table 5) clearly showed that the amount of Portland tended to reduce with an increase in the sawdust level, where the values dropped from 43.1 to 41.3, 36.4, 18.8, and 14.9% with the rise in sawdust levels from 0 to 25, 50, 75, and 100%, respectively. An analogous tendency was found with the calcite peak, and all concrete mixtures containing sawdust show a lower amount calcite (1.5–1.1%) compared to the control sample (2.2%). It is well known that the OPC amount in the concrete matrix plays a significant role in the production of the C-S-H gel, where a decrease in the OPC level and calcite can lead to weak bonds in the concrete matrix, thereby revealing lower strength values with the increase in sawdust content. Meanwhile, the high absorption and water demand of sawdust directly affected the geopolymerazition and dissolved the silicate and a showed a lower strength [54].

### 4.6. Predicted Compressive Strength of Proposed Concrete

ATLAB software was used to develop the proposed ANN model. Specifically, the best neuron number in the hidden layers was found to be fourteen during the iteration process. This result led to achieving the targeted performance of 0.01 related to the learning rate of 0.2. The measured data were divided into three parts—namely, training, testing, and validation. The ANN training with 124 tests was taken into account, while 14 and 23 tests were utilized for the validation and testing steps, receptively. The ANN training proceeded until it minimized the correlation value, in which a correlation value of 0.991 was achieved, as shown in Figure 13a. Whereas, the correlation value of the testing results is 0. 9878 (Figure 13b). Moreover, the mean error was considered to evaluate the efficiency of the network during the training and testing steps. In particular, the mean error (M) for the training data was noticed as 1.377, as shown in Table 6. These indicators values predicted the experimental data well, where the predicted CS was extremely close to the measured one.

Figure 14 compares the experimental and predicted results against molarity, GBFS to FA ratio, NS/NH ratio, solution to binder ratio, and time. The ANN model could realistically predict the actual CS of the alkali-activated concrete. This outcome proved that the ANN model results are consistent with experimental results. In addition, it was inferred the influence of each parameter on the CS of the concrete was clearly seen. For example, Figure 14a clearly shows the dependence of the amount of alkali-activated concrete strength enhancement on increased urea concentration. Both the experimental and predicted results shows a high degree of similarity. The enhancement of concrete strength is gradually increased with the increase in molarity. However, beyond a molarity value of 2, a not significant concrete strength enhancement is observed compared to the cost effect. As such. the molarity of 2 was kept and considered as 2 for further works. Similarly, with an increase in the NS/NH, the strength improved, as shown in Figure 14b. In the same regard, the contribution of fly ash concentration to the concrete strength enhancement was not significant, as shown in Figure 14c. In contrast, with the increase in the GBFS to FA ratio, the CS of the LWCs was improved. The optimum improvement of strength was achieved at an NS/NH ratio of 70/30. In a similar manner, the optimum value of the solution to binder ratio was 0.4 (Figure 14d).

### 4.7. Water Absorption (WA)

Figure 15 displays the effect of waste sawdust on the WA capacity of the alkali-activated concrete mixes at the curing age of 28 days. The WA of the specimens was elevated as the content of sawdust was increased in the concrete matrix. An increasing sawdust level from 0 to 25, 50, 75, and 100% as a substitute for natural aggregates could improve the WA by 9.7, 10.1, 13.4, 15.2, and 16.9%, respectively. At each level of sawdust replacement, the outcome of WA was affected considerably by the sawdust to river sand/crushed gravel ratio. At a 25% sawdust level, the water absorption was increased by 4.1% and this ratio increased with an increasing sawdust replacement and was recorded as more than 74% with 100% sawdust. As discussed in Section 3.3, with the rising contents of the sawdust the water demand was increased, thereby increasing the quantity of non-reacted silica and the structural porosity. Concretes containing higher amounts of sawdust showed an improved WA, which was due to the gel formation within the binder matrix. Ahmed et al. [6] attributed the high WA of concrete containing waste sawdust to its porosity and the presence of continuous channels. The other reason for such an enhanced WA ability may primarily be because of the availability of a high amount of free water that formed capillaries in concretes made from bottom ash, as demonstrated by Andrade et al. [55].

### 4.8. Drying Shrinkage (DS)

Figure 16 illustrates the test age-dependent variation in the DS values of alkali-activated concrete prepared using different levels of sawdust as a substitute for natural aggregates. The DS readings was taken at 3, 7, 14, 21, 28, 56, and 90 days and it was found that the value of DS for all samples was improved with the increase in the curing ages. The inclusion of sawdust waste into the alkali-activated concretes could in fact enhance the DS values, particularly at early ages. Furthermore, the values of DS were reduced with the rise in the sawdust levels, which was due to the specialty of the sawdust’s microstructures. For all the prepared concrete mixtures, it was observed that the increment in the drying shrinkage reduced with time; after 7 days of curing age, a more than 40% increment in the reading of drying shrinkage was recorded compared to the results of 3 days. However, this percentage was reduced with time and less than 6% and 2% were recorded after 28 and 90 days, respectively. At the early age of 3 days, the addition of sawdust as a natural aggregate replacement contributed to reducing the drying shrinkage by 2.1, 3.7, 6.2, and 7.1% with 25, 50, 75, and 100%, respectively. A comparable tendency was found after the curing age of 28 days, and the inclusion of sawdust showed more efficiency in reducing the shrinkage value by 3.5, 4.8, 7.1, and 7.6%, respectively with 25, 50, 75, and 100% compared to the control sample. The observed reduction in the DS of sawdust was ascribed to the inner curing effects of sawdust, providing some extra moisture and thus improvement in the DS response of the resulting mixes [56]. Actually, sawdust channels played a vital role by retaining water inside the alkaline solution-activated matrix, which in turn rendered the compensation of the essential moisture contents for the concrete matrix, leading to DS deformation afterward. Comparable results were obtained by Juarez et al. [57] and Tong et al. [58] on fiber-blended cement-based systems.

### 4.9. Sound Absorption

The acoustic absorption coefficients characterize the capacity of materials to absorb sound energy. Figure 17 shows the effect of the sawdust waste content on the measured sound absorption of the proposed concrete. All the specimens were tested under a frequency between 0 and 5000 Hz. In general, the tested specimens showed a better performance under frequency in a range between 500 and 3000 Hz and tended to absorb a higher amount of sound energy. The acoustic properties of the proposed concretes were improved with the inclusion of sawdust waste. The sound absorption coefficients tended to increase from 0.43 to 0.74 with the increase in the sawdust substitution levels from 0% to 100%, respectively. Figure 18 illustrates the noise reduction coefficient of the LWCs prepared with different amounts of sawdust wastes as a natural aggregate replacement. The results indicated that the rising level of sawdust waste substitution from 0% to 100% led to enhancing the acoustic properties of the concrete and increased the noise reduction coefficient from 0.124 to 0.237, respectively (Table 7). Several factors are responsible for the improved acoustic absorption performance. The increasing amount of sawdust wastes created more interconnecting voids at varied length scales within the alkali-activated concrete matrix and thus enhanced the noise reduction coefficients [59,60,61,62]. These porous materials showed an improved sound absorption in the high-frequency region, indicating a shift in the sound absorption frequency towards higher values with the increase in the sawdust content in the concrete. The observed decrease in the concrete density caused the frequency to shift towards higher values. In fact, sawdust, being a highly porous substance, can enhance the porosity of alkali-activated pastes when incorporated into the concrete matrix. The enhancement of the noise reduction coefficient with the addition of the sawdust aggregates was due to the air content and porosity increase in the concrete prepared with higher concentrations of sawdust aggregate. In previous studies [62,63], it was found that the sound absorption highly influenced the porosity and density of materials, and the noise reduction coefficient tended to increase with a decrease in the density of materials. The sound absorption of the porous materials is due to the energy loss by the friction generated in the walls of porous structures [64]. Consequently, the concrete specimen with a fraction of the voids generally showed higher values of sound absorption coefficients in the entire frequency domain [65].

### 4.10. Thermal Conductivity

Generally, alkali-activated LWCs are used in building partitioning. Thus, it is important to evaluate their thermal characteristics when applied in the external walls. Two methods were used to assess the thermal properties of the proposed concretes. First, the influence of the sawdust inclusion on the thermal conductivity of the concretes was measured by calculating the time of heat transfer at 28 days of curing age (Figure 19). The test results of thermal conductivity for the control and sawdust concretes showed a decrease with the rise in sawdust levels as a partial substitution for fine/coarse aggregates. The sample made with highest sawdust content (100%) showed the maximum time for heat transfer (188 min) compared to the control specimen (0% sawdust), which was only 36 min, while the other three mixtures containing 25, 50, and 75% sawdust displayed 61-, 108-, and 149-minute increases in heat transfer time, respectively. Second, the obtained values of the thermal conductivity coefficient were used to evaluate the thermal characteristics of the suggested lightweight concrete. The results presented in Figure 20 make it clear that the increasing sawdust content as a natural aggregate replacement resulted in an enhancement in the thermal properties of the proposed concrete. For all the tested specimens, the value of the thermal conductivity coefficient trend to decrease with the increasing sawdust waste content. It was found the thermal conductivity coefficient k-value dropped from 0.39 to 0.24, 0.19, 0.13, and 0.09 W/m.K with an increasing level of replacement from 0 to 25, 50, 75, and 100%, respectively. In study by Liu et al. [65], it was reported that the enhancement in thermal properties tended to increase with an increase in the porosity of specimens and a reduction in the density. This indicated that the inclusion of sawdust waste led to a reduction in the density of the proposed concrete as well as an increase in the total porosity, thereby lowering the thermal conductivity. Because of its light weight and porosity, sawdust with a low density can reduce the thermal conductivity of formulated concretes. It was acknowledged that [66] this decrease in the thermal conductivity may be related to the convection process, wherein the pore density, distribution, and geometry within the concrete matrix play a significant role. Figure 21 shows the thermal conductivity and density correlation of all the prepared concretes with the obtained correlation coefficient of 0.9048. The lightweight aggregates concretes are known to reveal a tendency of decreasing density, resulting in more porosity and hence a reduction in the thermal conductivity [67,68].

### 4.11. Environmental and Economic Benefits

The total CO_2_ release, cost, and energy utilization of natural aggregates, including river sand, crushed gravel, and fine and coarse sawdust, were calculated to determine the life cycle of every material (Table 4 in Section 3.7). The results indicated that the river sand and crushed gravel required a higher amount of diesel during the synthesis than sawdust waste, leading to a rise in the CO_2_ release, cost, and energy usage. Natural aggregates consumed a higher amount of energy in the range of 0.134–0.148 GJ/m^3^ compared to fine (0.018 GJ/m^3^) and coarse (0.021 GJ/m^3^) sawdust. This showed direct proportionality to the energy expenditure, CO_2_ release, and enlarged outlay of natural aggregates. It revealed a higher carbon dioxide release (0.012 tonne/m^3^) than sawdust (0.0008 tonne/tonne). Similar to the CO_2_ release, the cost involvement for river sand and crushed gravel were the highest from sawdust. This was mainly because of the collection and preparation stages, such as crushing and sieving, as well as the longer transportation distance. This, in turn, raised the cost of river sand and crushed gravel to 55 and 65 RM/m^3^ compared to the respective fine and coarse sawdust cost of 34.5 and 36 RM/m^3^. The replacement of natural aggregates by lightweight fine and coarse sawdust in the alkaline-activated matrix was shown to be essential to attain the sustainability conditions such as lower CO_2_ release, cost, and energy utilization.

Figure 22 depicts the carbon dioxide emissions of the prepared concretes against different contents of sawdust. The impact of the sawdust substitution for river sand and crushed aggregates on the CO_2_ release of the lightweight alkaline solution-activated concretes was studied. The CO_2_ release was observed to drop from 10.9 to 8.3, 5.8, 3.3, and 0.8 kg/m^3^ with the rise in the sawdust contents from 0 to 25, 50, 75, and 100%, respectively. The alkaline solution-activated concrete matrix containing 100% sawdust (1 kg/m^3^) could reduce the CO_2_ release by 90% compared to 10.9 kg/m^3^ for natural aggregates. This lowering in the CO_2_ release from alkaline solution-activated concretes containing sawdust as an alternative to natural aggregates affirmed the possibility of making a simple sustainable development in the construction sector.

Figure 23 shows the effect of sawdust replacing river sand and crushed gravel on the aggregate cost calculation of the prepared concrete. The usage of sawdust at a high level (100%) as a substitute for sand and gravel also saved money. The price of material by weight depended on the life cycle (Table 4), which showed a direct influence on the final price of the concrete mixes made of aggregates. Furthermore, the cost of aggregates was reduced from 62.3 to 55.8, 49.4, 42.9, and 36.6 RM/m^3^ with the rise in the sawdust contents as a substitute to natural aggregates at 0 to 25, 50, 75, and 100%, respectively. It was shown that, by implementing sawdust as an alternative to river sand and crushed gravel, a sustainable concrete can be achieved.

Figure 24 displays the energy consumption of the prepared concrete against the various sawdust contents. Depending on the life cycle and energy use of every substance, the total energy expenditure of every alkaline solution-activated concrete was evaluated. The energy utilization of the produced concretes was reduced with the addition of sawdust as a substitute for river sand/crushed gravel. In comparison to the energy use for the 0.15 GJ/m^3^ concrete with a 0% sawdust content, the values were reduced to 0.11, 0.08, 0.05, and 0.022 GJ/m^3^ for mixes made with sawdust levels of 25, 50, 75, and 100%, respectively. However, all the alkaline solution-activated mixes with the maximum quantity of sawdust (100%) required much lower energy than the one made with natural aggregates (0.15 GJ/m^3^). A low diesel utilization during the life cycle of sawdust can directly influence the final energy use of the proposed concretes. The low price, CO_2_ release, and energy utilization by the produced sawdust wastes were the primary factors that allowed attaining the desired sustainability of the alkaline solution-activated aggregates.

## 5. Conclusions

The conclusions of this study are as follows:(1)Replacing the natural aggregates, including river sand and gravel, with sawdust wastes could influence the flow ability, setting time, and CS development. The slump of the prepared concrete dropped from 130 to 74 mm with the rise in the corresponding sawdust levels from 0% to 100%. Both the initial and final setting times were shorter than those of the control sample for the mixture containing 100% sawdust.(2)The increment in the sawdust content from 0% to 100% negatively affects the strength gain. Meanwhile, the concretes prepared with 100% sawdust as a substitute for natural aggregates attained a satisfactorily high CS (48.6 MPa), allowing the proposed formulation to be suitable for diverse applications in the construction sector as a high-performance lightweight cement-free concrete.(3)The ANN provided satisfactorily results for estimating the mechanical properties of AAMs compared to the ANN combined with the genetic algorithm and multiple linear regression models.(4)Replacing the natural aggregates with sawdust led to a decrease in the drying shrinkage of the tested specimens.(5)All the mixes prepared with sawdust as a substitute for natural fine/coarse aggregates exposed to elevated temperatures displayed a lower resistance and higher strength loss.(6)The present findings proved that these new lightweight alkali solution-activated concretes are more environmentally friendly compared to conventional aggregate-based concrete. The replacement of sand and gravel aggregates with sawdust in the concrete matrices could produce a more efficient product with a lesser CO_2_ release, lower cost, and lower fuel use compared to one using natural aggregates.(7)To reduce the cost of the alkaline solution-activated aggregates by 41.2% (RM 36.6) as opposed to RM 62.3 for natural aggregates, it is suggested to employ 100% sawdust as a substitute for river sand and crushed gravel. Thus, it is feasible to achieve a new sustainable concrete without using natural aggregates, which is effective for sustainable development. Furthermore, the proposed new formulation will remarkably reduce the CO_2_ release below the 85% fuel production obtained using natural aggregates.(8)Apart from the environmental benefits, the designed concretes may offer a better-quality product with distinct mechanical characteristics, which is useful for several companies interested in lightweight concrete production. Briefly, sawdust wastes as an alternative to natural aggregates can offer several benefits and serve to accomplish the goal of sustainability in civil engineering construction with environmental friendliness.

## Figures and Tables

**Figure 1 materials-13-05490-f001:**
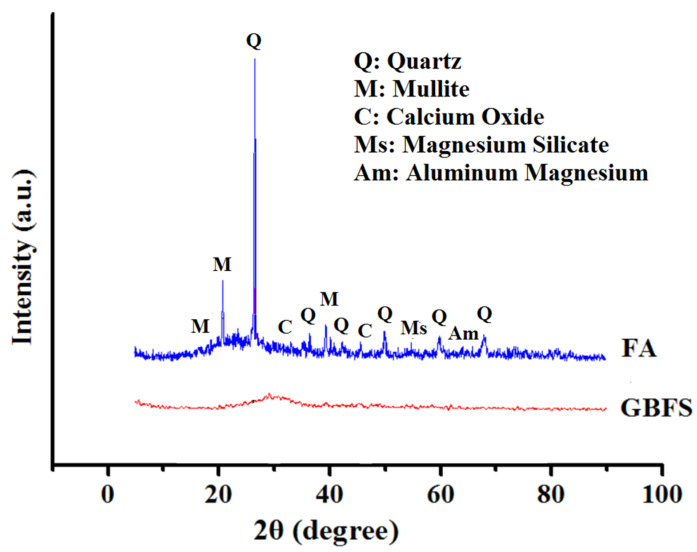
XRD diffractograms of fly ash (FA) and slag (GBFS).

**Figure 2 materials-13-05490-f002:**
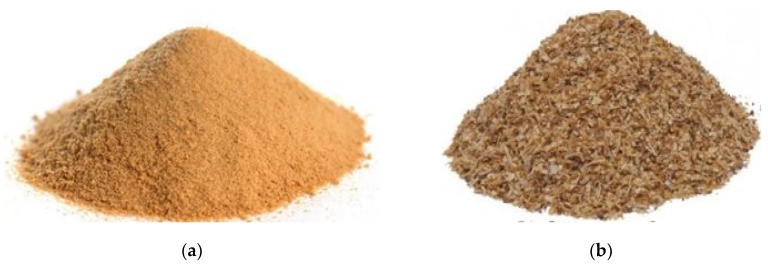
(**a**) Fine sawdust, (**b**) coarse sawdust.

**Figure 3 materials-13-05490-f003:**
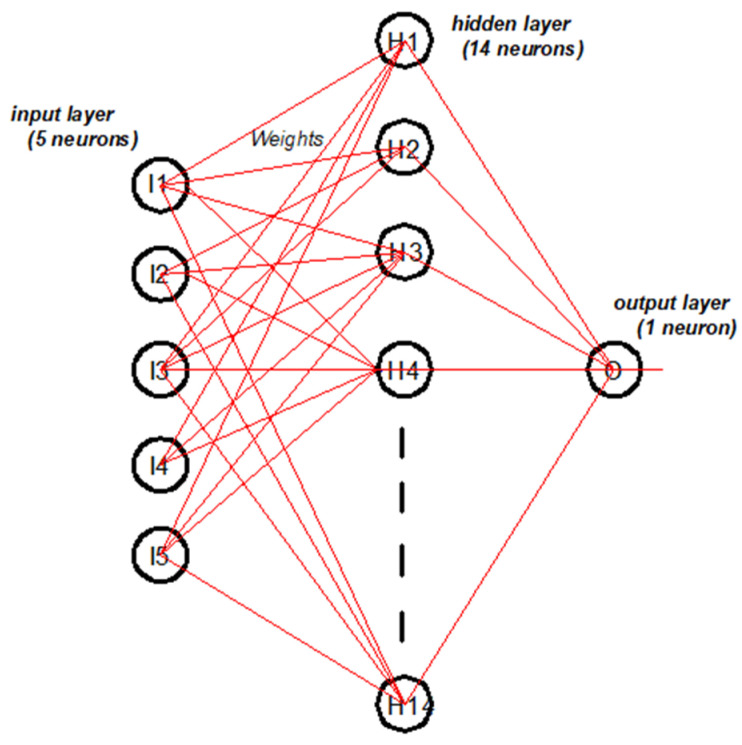
Artificial neural network processing.

**Figure 4 materials-13-05490-f004:**
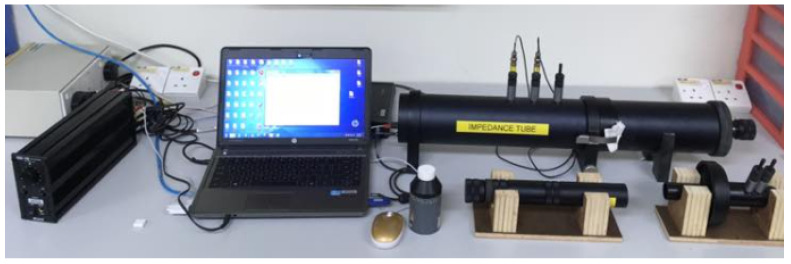
Impedance tube instrument.

**Figure 5 materials-13-05490-f005:**
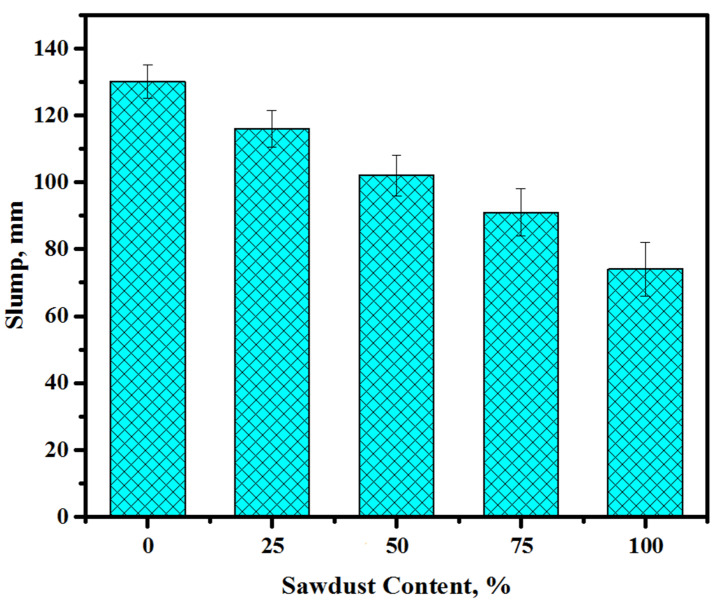
Slump values of the prepared alkali-activated concretes.

**Figure 6 materials-13-05490-f006:**
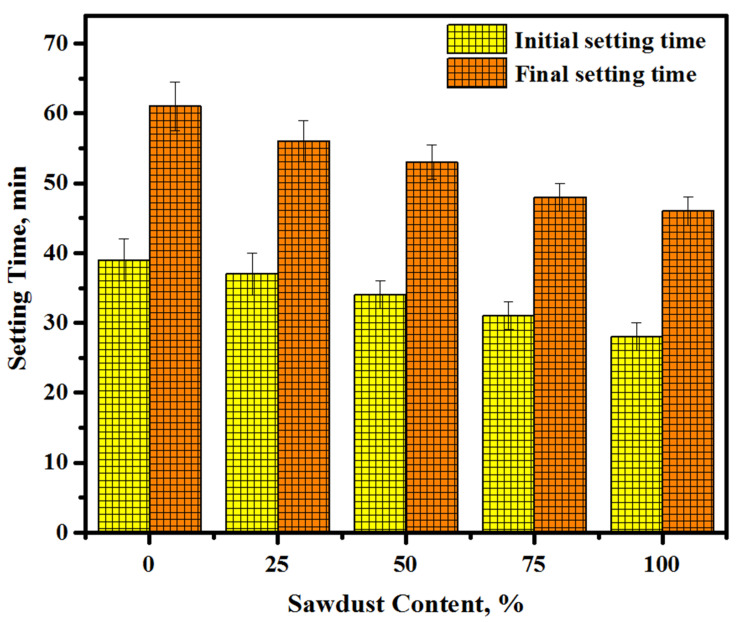
Effect of the sawdust content on the concretes’ initial and final setting times.

**Figure 7 materials-13-05490-f007:**
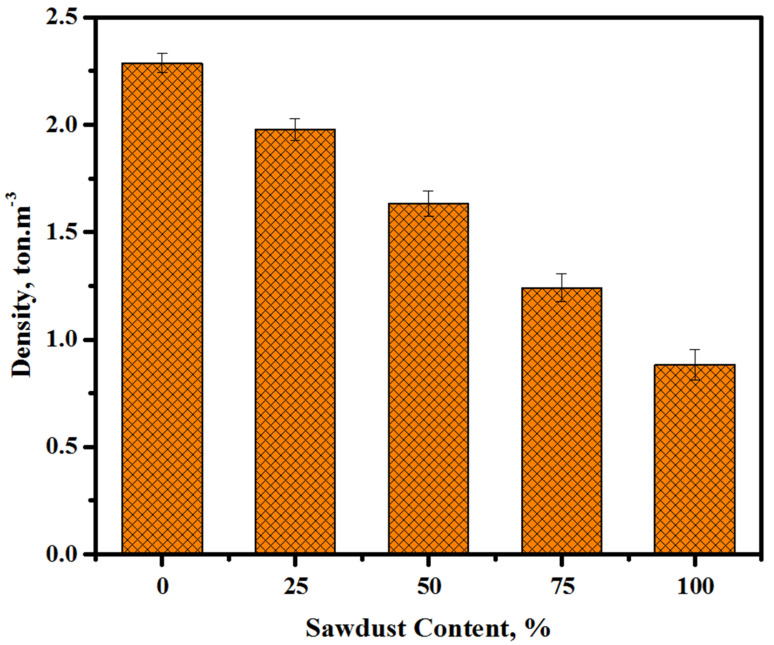
Densities of the prepared concretes with various sawdust contents.

**Figure 8 materials-13-05490-f008:**
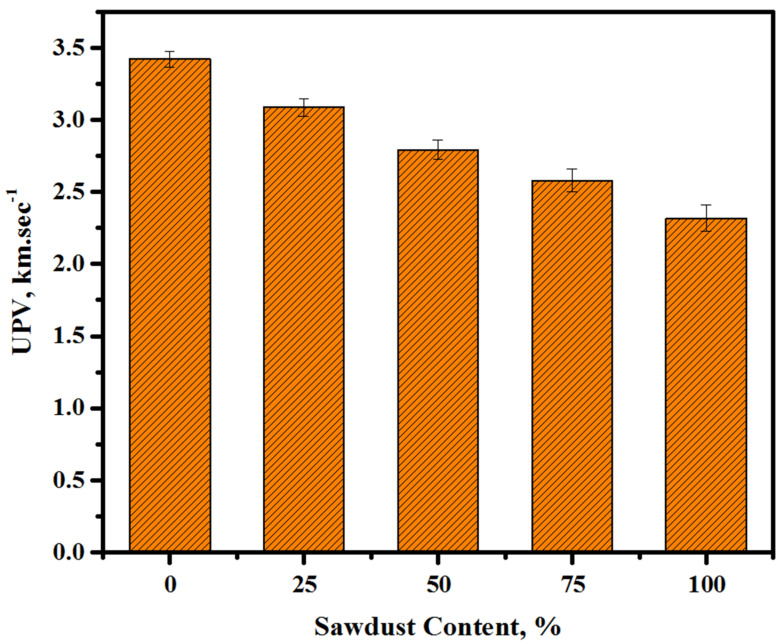
UPV readings of the prepared concretes with various sawdust contents at 28 days of age.

**Figure 9 materials-13-05490-f009:**
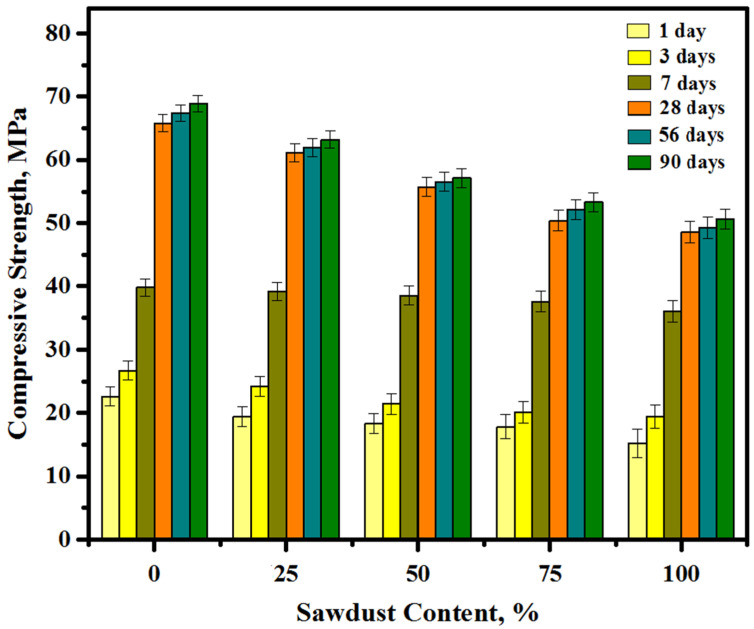
Compressive strength results of the prepared concrete at various sawdust contents.

**Figure 10 materials-13-05490-f010:**
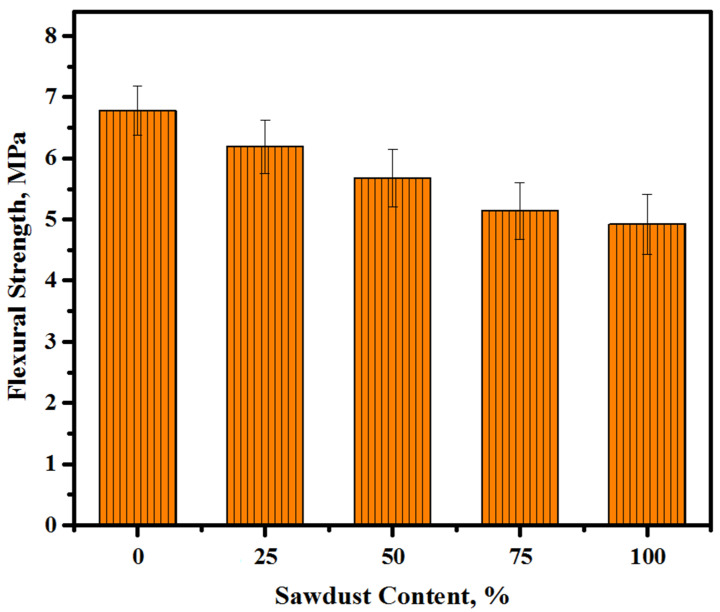
The FS of the prepared LWCs containing various amounts of sawdust.

**Figure 11 materials-13-05490-f011:**
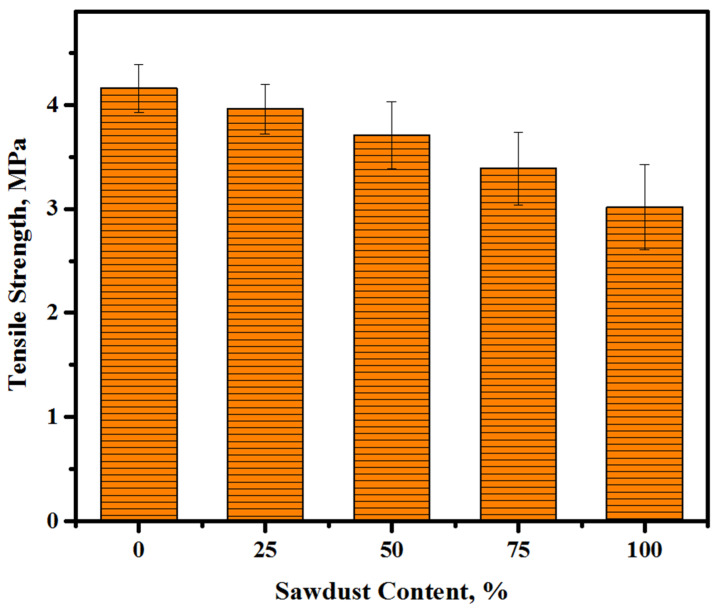
The STS of the LWCs against the sawdust contents.

**Figure 12 materials-13-05490-f012:**
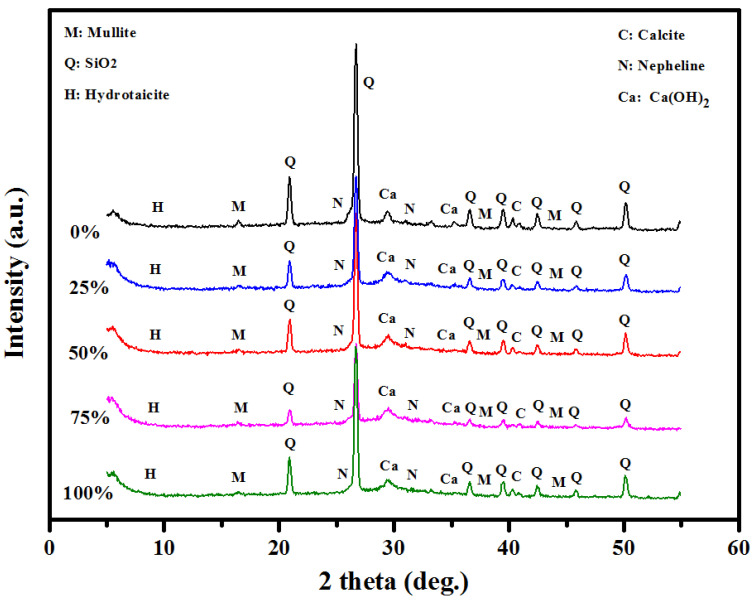
XRD analysis of the prepared concrete with various sawdust contents.

**Figure 13 materials-13-05490-f013:**
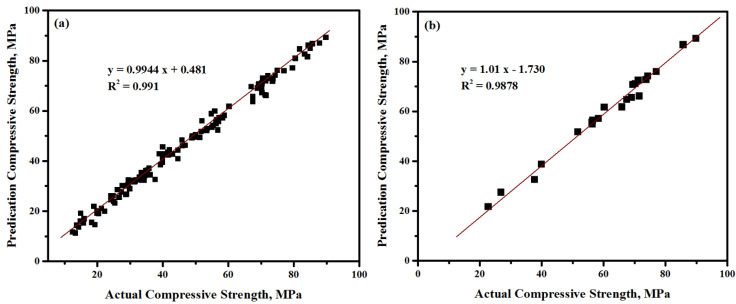
Predicted and measured CS correlation of the proposed LWCs for the (**a**) training and (**b**) testing data.

**Figure 14 materials-13-05490-f014:**
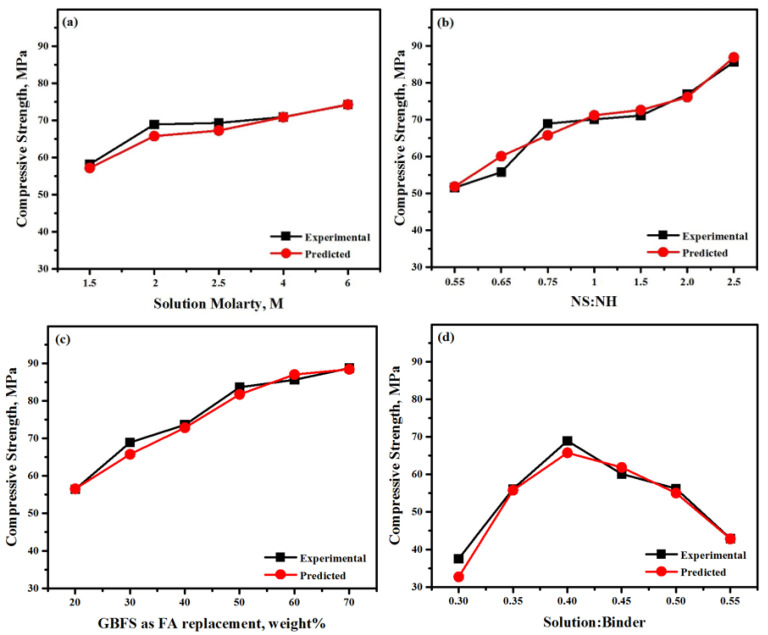
Predicted and actual evolution of alkali-activated strength as a function of (**a**) molarity, (**b**) NS/NH, (**c**) FA/GBFS, (**d**) and solution to binder ratio.

**Figure 15 materials-13-05490-f015:**
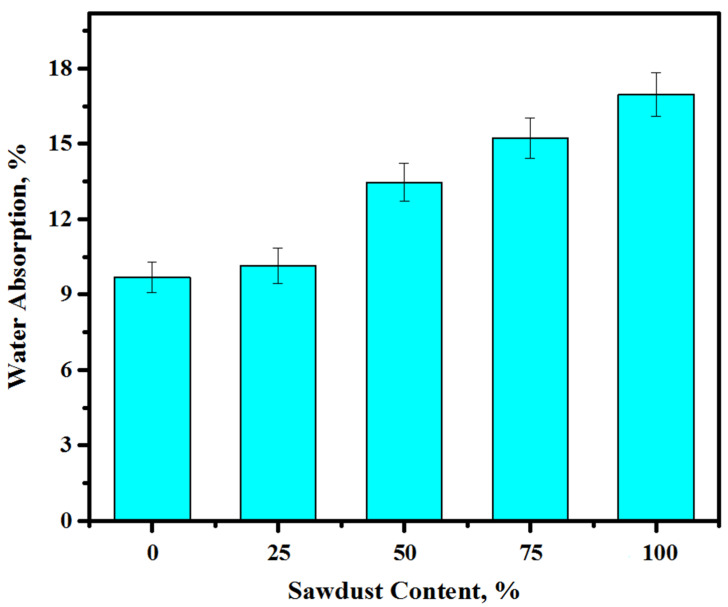
The water absorption of the prepared concrete against different sawdust contents.

**Figure 16 materials-13-05490-f016:**
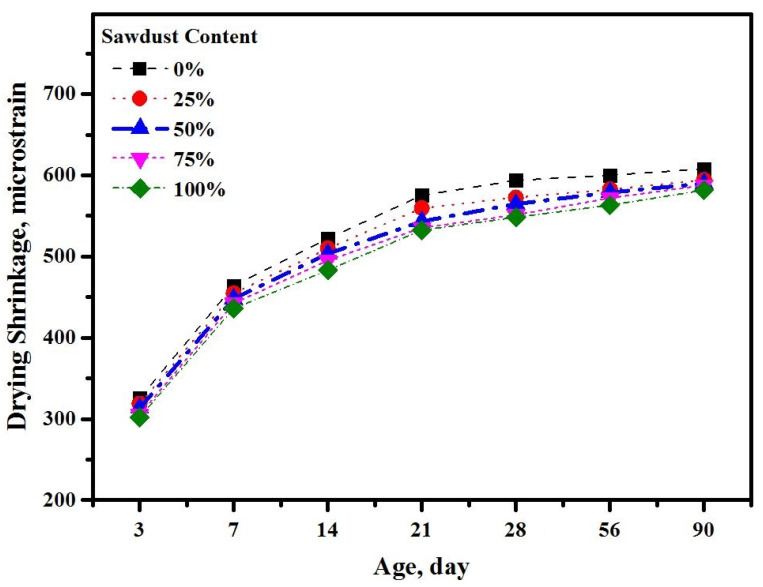
Drying shrinkage of the prepared concrete with various sawdust contents.

**Figure 17 materials-13-05490-f017:**
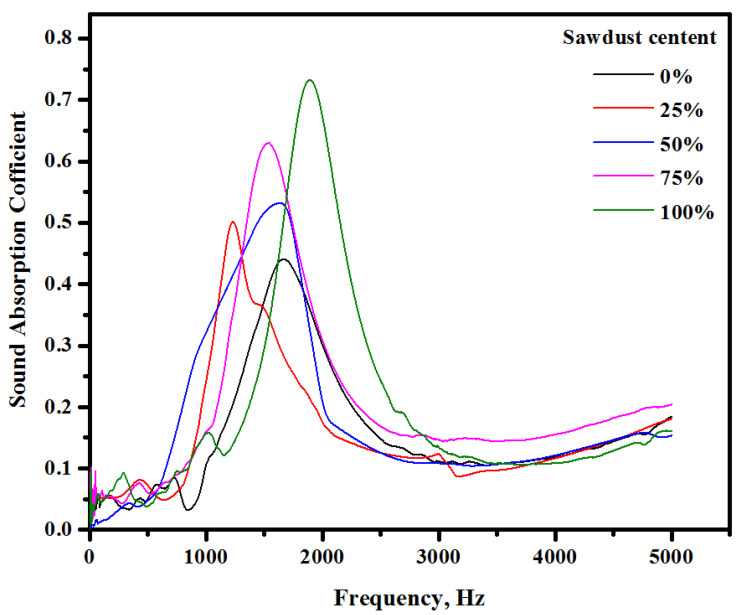
Sound absorption coefficients of the concretes prepared with different amounts of sawdust wastes.

**Figure 18 materials-13-05490-f018:**
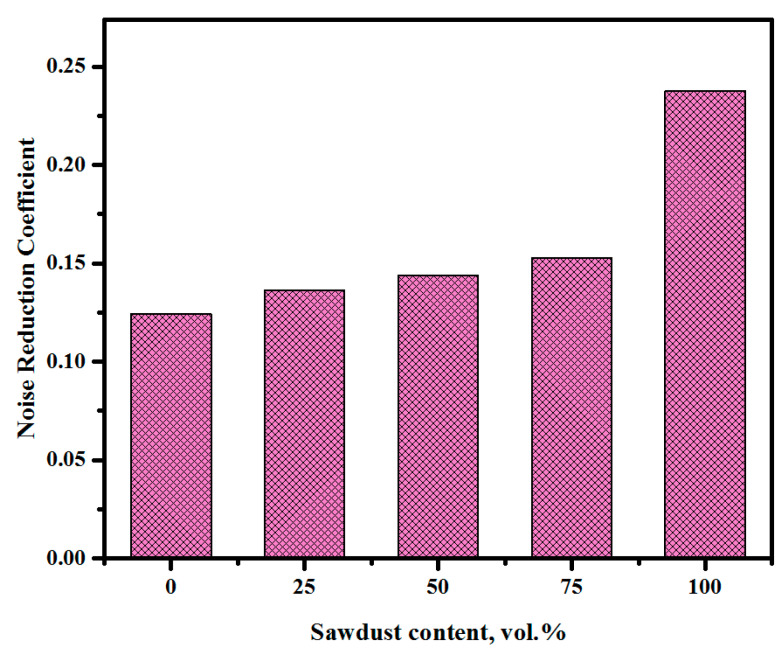
Effects of the sawdust contents on the noise reduction coefficient of the proposed alkali-activated concretes.

**Figure 19 materials-13-05490-f019:**
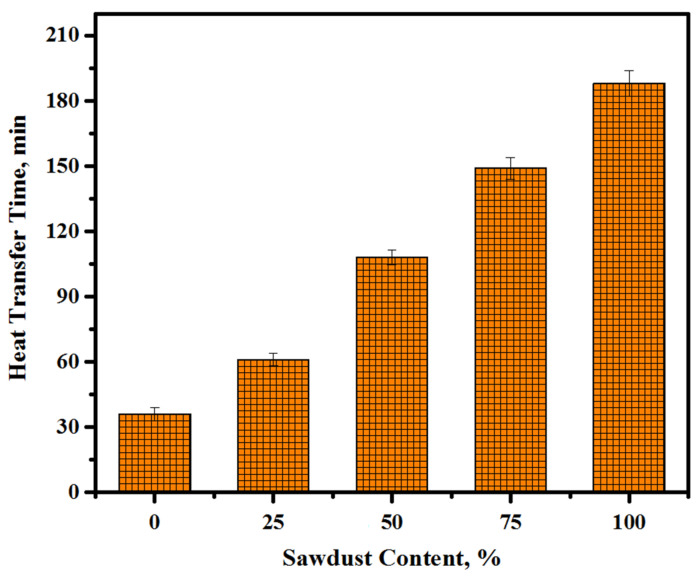
Thermal conductivity of the prepared concrete at various sawdust contents.

**Figure 20 materials-13-05490-f020:**
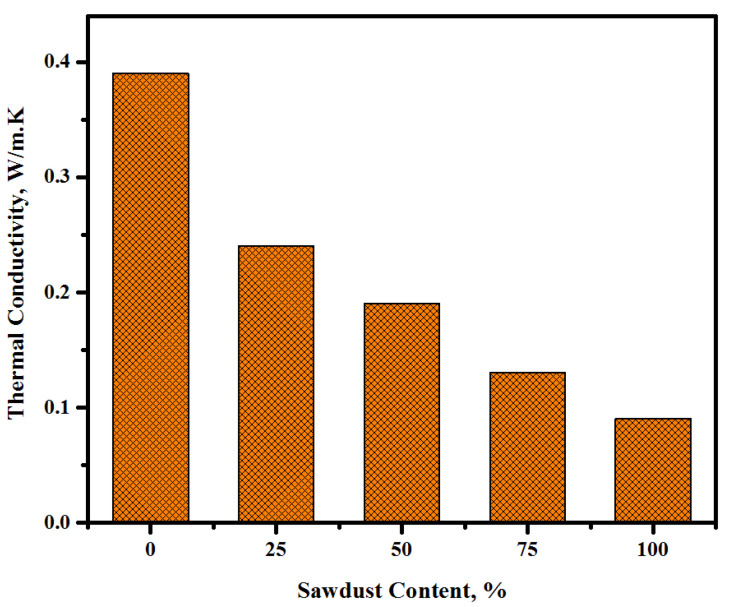
Thermal conductivity coefficients of the prepared concretes at various sawdust contents.

**Figure 21 materials-13-05490-f021:**
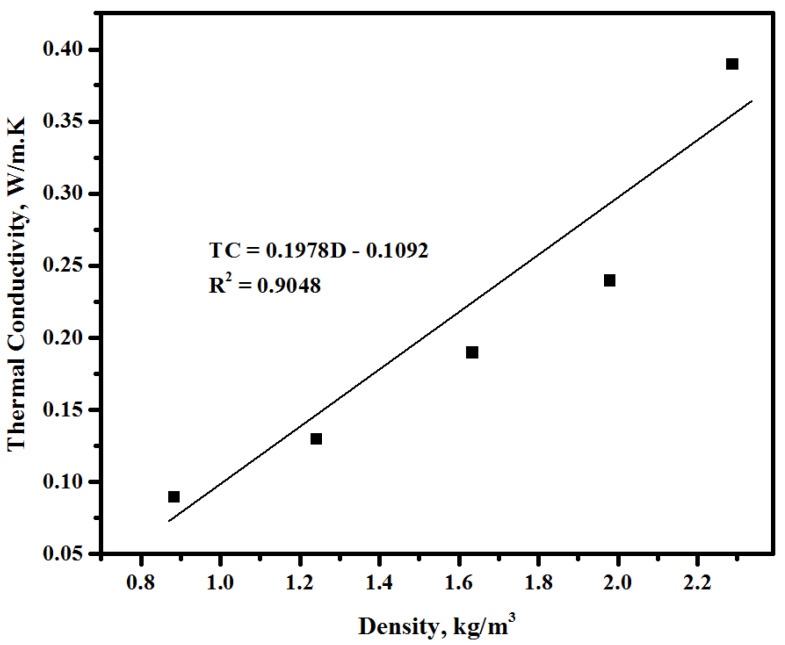
Relationship between the heat transfer time and the density of the prepared concretes.

**Figure 22 materials-13-05490-f022:**
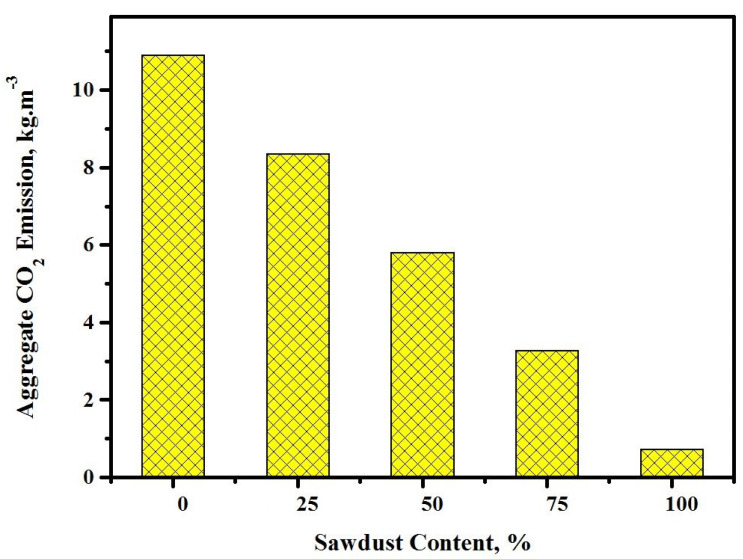
Effect of the sawdust content on the prepared concrete aggregate carbon dioxide emission.

**Figure 23 materials-13-05490-f023:**
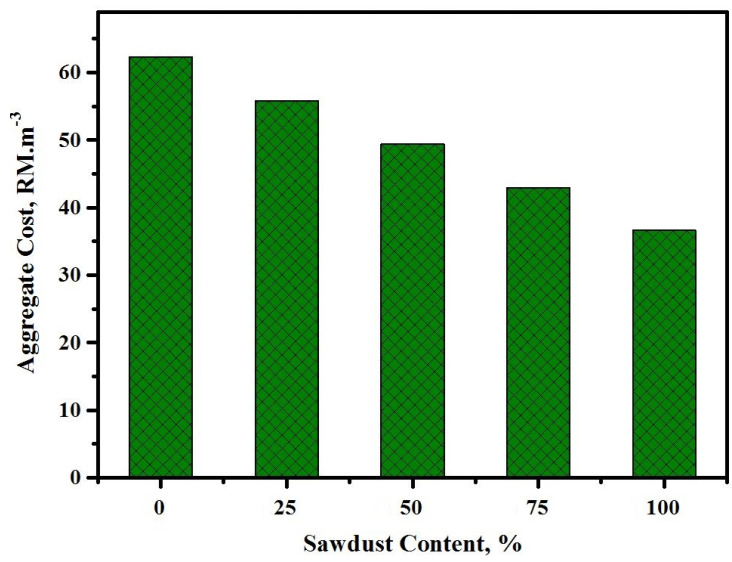
Effect of the sawdust content on the prepared concrete aggregate cost.

**Figure 24 materials-13-05490-f024:**
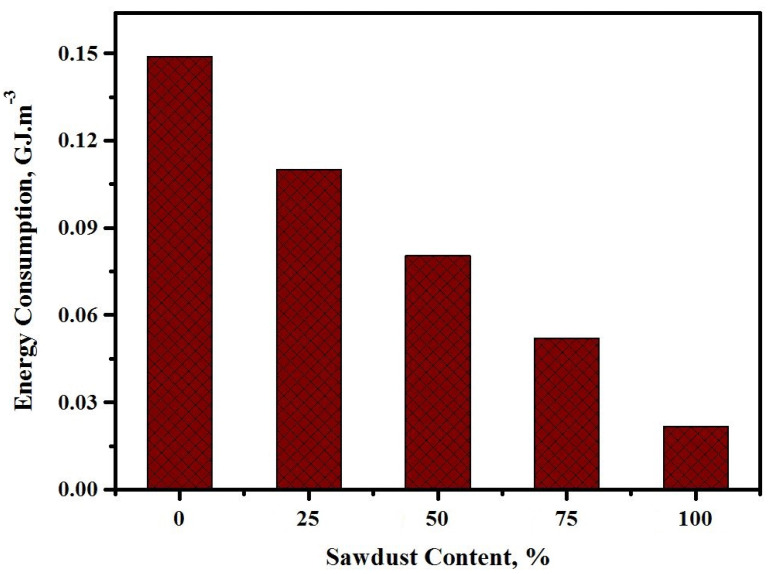
Energy consumption of the concretes prepared at various sawdust contents.

**Table 1 materials-13-05490-t001:** Chemical constituents (in %) of the waste sawdust.

Chemical Compositions (%)						
**Cellulose**	**Al_2_O_3_**	**Fe_2_O_3_**	**CaO**	**MgO**	**K_2_O**	LOI
87.0	2.5	2.0	3.50	0.23	0.01	4.76

**Table 2 materials-13-05490-t002:** Mix design of the proposed lightweight alkali-activated concrete (Kg/m^3^).

Mix	Binder (kg/m^3^)		Solution (kg/m^3^)		Fine and Coarse Aggregates (kg/m^3^)			
	FA	GBFS	NH	NS	River Sand	Crushed Gravel	Fine Sawdust	Coarse Sawdust
S0	315	135	104	78	845	950	0	0
S25					634	712	22	26
S50					422	475	45	47
S75					211	237	67	71
S100					0	0	90	95

**Table 3 materials-13-05490-t003:** Machine and material information for the life cycle calculation.

Item	Value
Speed of truck, km/h	80
Diesel cost, L/km	0.09
Diesel cost, RM/L	2.18
Truck capacity, m^3^	12
Transportation cost of 1 m^3^, RM/km	0.75
Density natural coarse aggregate, kg/m^3^	1820
Density of river sand, kg/m^3^	1640
Fine sawdust density, kg/m^3^	176
Coarse sawdust density, kg/m^3^	182
CO_2_ release for 1 L diesel, ton	0.0027
Energy cost for 1 L diesel, GJ	0.0384

**Table 4 materials-13-05490-t004:** The CO_2_ release, expenditure, and energy use of every material depended on life cycle.

Material	CO_2_ Emission, ton/m^3^	Cost, RM/m^3^	Energy Consumption, GJ/m^3^
Sand	0.009	55	0.134
Gravel	0.012	65	0.148
Fine sawdust	0.0006	34.5	0.018
Coarse sawdust	0.0008	36	0.021

**Table 5 materials-13-05490-t005:** XRD peak amount by wt.%.

Index	Sawdust Content by Vol%				
	0%	25%	50%	75%	100%
Quartz, SiO_2_	52.1	54.8	60.2	78.4	81.8
Portland, Ca(OH)_2_	43.1	41.3	36.4	18.8	14.9
Calcite, CaCO_3_	2.2	1.5	1.3	1.1	1.2
Others	2.6	2.4	2.1	1.8	2.1

**Table 6 materials-13-05490-t006:** The comparison between the experimental data for the testing set and the predicted results from the ANN model.

Experimental no.	Time (Day)	GBFS/FA	Solution/Binder	Molarity (M)	NS/NH	Actual (MPa)	Predicted (MPa)	Error	Absolute Error
1	1	30	40	2	0.75	22.60	21.87	−0.73	0.73
2	3	30	40	2	0.75	26.70	27.61	0.91	0.91
3	7	30	40	2	0.75	39.80	38.83	−0.97	0.97
4	28	30	40	2	0.75	65.80	61.87	−3.93	3.93
5	56	30	40	2	0.75	67.40	64.90	−2.50	2.50
6	90	30	40	2	0.75	68.90	65.75	−3.15	3.15
7	90	40	40	2	0.75	73.60	72.82	−0.78	0.78
8	90	60	40	2	0.75	85.60	87.00	1.40	1.40
9	90	70	40	2	0.75	89.70	89.38	−0.32	0.32
10	90	20	40	2	0.75	56.40	56.55	0.15	0.15
11	90	30	30	2	0.75	37.60	32.76	−4.84	4.84
12	90	30	35	2	0.75	56.10	55.78	-0.32	0.32
13	90	30	45	2	0.75	60.10	61.85	1.75	1.75
14	90	30	50	2	0.75	56.20	55.02	−1.18	1.18
15	90	30	40	6	0.75	74.20	74.26	0.06	0.06
16	90	30	40	4	0.75	71.40	66.26	−5.14	5.14
17	90	30	40	2.5	0.75	69.30	70.86	1.56	1.56
18	90	30	40	1.5	0.75	58.20	57.22	−0.98	0.98
19	90	30	40	2	2.5	85.60	86.86	1.26	1.26
20	90	30	40	2	2	76.90	76.08	-0.82	0.82
21	90	30	40	2	1.5	71.10	72.61	1.51	1.51
22	90	30	40	2	1	70.10	71.18	1.08	1.08
23	90	30	40	2	0.75	51.60	51.88	0.28	0.28
Sum									35.62
Mean									1.37

**Table 7 materials-13-05490-t007:** Effect of the sawdust content on the noise reduction coefficient (NRC).

Frequency (Hz)	0%	25%	50%	75%	100%
250	0.040087632	0.053833563	0.046065998	0.03238883	0.082436911
500	0.050464286	0.072491523	0.060480702	0.047957295	0.038381466
1000	0.105207679	0.241543934	0.159514703	0.320905945	0.156701336
2000	0.301440476	0.176171063	0.308779167	0.20956316	0.673137931
NRC	0.124300018	0.136010021	0.143710142	0.152703808	0.237664411

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
