# Peer review of "Engineering Properties of Waste Sawdust-Based Lightweight Alkali-Activated Concrete: Experimental Assessment and Numerical Prediction"

_materials, 2020, doi:10.3390/ma13235490_

Round 1
Reviewer 1 Report
Extensive research of sawdust based lightweight alkali-activated concrete was presented. Results are presented clearly and understandably in most cases.
In Chapter 3, the experiments are described, however, I miss the description of drying shrinkage testing and the description of water absorption testing. Consequently, I cannot assess the results properly.
In Chapter 3.7, Environmental and economic benefits, I miss the comparison of CO2 emissions of cement based concrete and alcali activated concrete without cement. Comparison of CO2 emissions (cost, energy consumption) given in this paper is only about the location of recourses connected to specific places and does not allow a general conclusion. Table 2 is identical to Table 6.
Some Figures need improvement, should be presented in better graphical quality.
Although this paper is written in understandable English, the text needs language improvement.
I recommend to considere a change of article title. Current title
Prediction of engineering properties of wastes sawdust based lightweight alkali-activated concrete: Experimental and numerical assessment
does not clearly capture the content. Suggestion:
Engineering properties of waste sawdust based lightweight alkali-activated concrete: Experimental assessment and numerical prediction
Author Response
As attached

Reviewer 2 Report
The paper is very interesting, since it carries out an extensive experimental study on the behavior of concrete with waste sawdust, in order to achieve a lighter and more sustainable concrete.
The work is well structured, has an adequate introduction with numerous bibliographical references and with numerous test that allow us to understand the behavior of this type of concrete.
For contributing to improve the article, I indicate some non-relevant aspects that would improve the work:
- Review the numbering of tables and figures, some are repeated.
- Figure 13, a) and b) in black, not red.
- It would be interesting to unify the figures a bit, in backgrounds and colors.
- Figure 18, the table within the figure can be improved.
- It would be interesting if the figures of the mechanical tests were included and other variables of these tests were valued.
Author Response
As attached

Reviewer 3 Report
1. L39-41. References [1] and [2] do not claim this description, essentially. (Please see the titles of [1] and [2].) In addition, wood waste is one of the biomass fuels that replace petroleum resources, and it is considered that there is a socially positive view as a recycling-oriented resource from the viewpoint of biodegradability. The rationality of this research should not be asserted from this perspective.
2. L 280. “3.7 Environmental and economic benefits.” This section is not a result but experimental details. However, the basis for the calculation is not shown, especially for LCA.
3. L 134. XRF can be evaluated qualitatively, but not quantitatively.
4. L164. Wood densities are too low. Kempas wood is a kind of heavy timber.
5. L 170. Table 1. Why is SiO2 as much as 87 % in the mass composition of wood?
6. L 321. What is the state of sawdust 100 %?
Author Response
As attached

Round 2
Reviewer 1 Report
The manuscript was improved according to suggestions.
Reviewer 3 Report
I could understand. The necessary corrections have been made.